# Dimensions of music use motivations: Genetic and environmental underpinnings, and associations with Big Five and Empathy traits

Heidi Marie Umbach Hansen[1,2,3]*, Espen Røysamb[1,3,4], Olav Mandt Vassend[1],
Nikolai Olavi Czajkowski[1,3,5], Tor Endestad[1,2,6], Jonna Katariina Vuoskoski[1,2,7],
Anne Danielsen[2,7], Bruno Laeng[1,2]

**1** Department of Psychology, University of Oslo, Oslo, Norway, **2** RITMO Centre for Interdisciplinary
Studies in Rhythm, Time and Motion, University of Oslo, Oslo, Norway, **3** Department of Psychology,
PROMENTA Research Center, University of Oslo, Oslo, Norway, **4** Division of Mental and Physical
Health, Norwegian Institute of Public Health, Oslo, Norway, **5** Department of Mental Health and Suicide,
Norwegian Institute of Public Health, Oslo, Norway, **6** Department of Neuropsychology, Helgeland
Hospital, Mosjøen, Norway, **7** Department of Musicology, University of Oslo, Oslo, Norway

* h.m.u.hansen@psykologi.uio.no

## Abstract

People differ in their motivation for seeking musical experiences and these differences appear to be partially attributable to their personality. However, little is known about the role of genetic and environmental factors in shaping individual differences in motivations for music use and their shared etiology with personality. This study investigated, using the classical twin design in a sample of 2611 Norwegian twins, the genetic and environmental architecture of four dimensions of motivations for music use (musical transcendence, emotion regulation, social bonding, and musical identity and expression). We also examined their phenotypic associations, as well as genetic and environmental overlap with general personality traits, including the Big Five and Trait Empathy facets. Additive genetic and unique environmental effects largely accounted for the variation in the four dimensions (heritability 38%−52%, mean = 45%), though univariate analyses of the social bonding dimension indicated common environmental influences for this specific subscale. In line with previous studies, trait-congruent phenotypic associations were found between specific personality traits and music use motives, including facets of negative emotionality and musical emotion regulation. Other facets – particularly aesthetic sensitivity (open-mindedness) and fantasy (trait empathy) – were closely tied to all dimensions. Multivariate biometric modeling of the links between personality traits and motivations for music use revealed that these relationships were primarily driven by correlated genetic factors, with personality traits accounting for 40%−66% of the total genetic variance and 5%−17% of the total environmental variance in the music use motivation dimensions. These findings shed new light on the etiology of dispositional music

**Data availability statement:** The data used in this study are drawn from the Norwegian Twin Registry (NTR) and cannot be shared publicly because of legal and ethical restrictions. The NTR is regulated under the Regulations on Population-Based Health Surveys. Applicants wishing to access data from the NTR must document a valid legal basis for the processing of personal data and pre-approval from the Regional Committees for Medical and Health Research Ethics (REK). Applicants who meet the criteria for access can submit an application form via helsedata.no. For more information about access procedures, please refer to the following resources: The Norwegian Twin Registry: https://www.fhi.no/hs/tvilling/tvillingregisteet/. The Norwegian Institute of Public Health: https://www.fhi.no/hd/datatilgang/.

**Funding:** This work was partially supported by the Research Council of Norway (https://www.forskningsradet.no/en/), project number 262762, 288083, and 314843. Data collection was funded by the University of Oslo (funding of research infrastructure, obtained by ER, NOC, and OMV). The funders had no role in study design, data collection and analysis, decision to publish, or preparation of the manuscript.

**Competing interests:** The authors have declared that no competing interests exist.

use and provide a comprehensive baseline for future investigations of associations with personality, mental health, and well-being.

## Introduction

Music is an immensely rich human experience, but people differ in their motivations for engaging with music. These can be numerous and multifaceted – some might primarily engage with music for immediate hedonic rewards or just to pass the time, while for others, the motivations may be more profound, as a need to foster social, emotional, or spiritual needs [1,2]. Different ways of engaging with music may, indirectly or directly, impact well-being [3–5] and the formation of aesthetic preferences [6]. Still, we know very little about the nature and origins of this interindividual variability in music use, though recent evidence points to considerable genetic influences on affective musical tendencies like musical sensibility [7] and music reward sensitivity [8].

Music use is also linked to personality traits, particularly those subsumed under the Five-Factor [9] and Big Five [10] models of personality, namely, extraversion, agreeableness, conscientiousness, negative emotionality (alternative labels include neuroticism or emotional stability), and open-mindedness (alternative labels include openness to experience, intellect, or imagination), hence referred to as the "Big Five" [11,12]. Specifically, there are well-established links between open-mindedness and cognitive uses of music, as well as negative emotionality and musical emotion regulation [13–17]. However, there is limited knowledge about the role of genetic and environmental factors in explaining these relationships, as well as which specific personality facets are most important. Moreover, an open question remains about the putative role of Trait Empathy, another key aspect of personality. Empathy has previously been linked with individual differences in music preferences, music reward sensitivity, and emotional responsivity to music [e.g., 18–21], suggesting that empathic tendencies might also be related to the motivational basis of music engagement. It is important to note, however, that there is considerable overlap between the two personality frameworks (i.e., the Big Five and Trait Empathy) [22,23]. Since few studies have considered the two jointly in relation to musical traits and thus accounted for their overlap, possible unique contributions need further investigation [but see, e.g., 24].

Here, we aimed to reduce these gaps in the literature by first examining the underlying genetic and environmental architecture of four empirically validated [25] motivations for music use, namely, musical transcendence, emotion regulation, social bonding, and musical identity and expression. Second, considering the previously identified links between individual patterns of music use and personality, we also sought to elucidate their phenotypic and potential etiological overlap with the Big Five and Trait Empathy.

### Conceptualizing and measuring motivations for music use

Scientific endeavors to describe the potential functions, benefits, or uses of music can take several forms. While some researchers have addressed this theoretically,

such as seeking to explicate the evolutionary origins of music [26,27], others have sidestepped evolutionary claims and approached this empirically, focusing on the more tractable patterns of everyday music use, like looking at individual's listening preferences and self-reported reasons for music listening [e.g., 2,15]. Combined, the two approaches have generated a particularly abundant – and somewhat overlapping – list of potential functions. However, as pointed out in a literature review by Schäfer and colleagues [1], while the number of distinct dimensions have varied across studies and theoretical frameworks, the existing literature could broadly be distilled down to a few functions, each associated with cognitive or self-related processes (e.g., escapism, intellectual, or eudaimonic experiences), emotional purposes (e.g., to increase positive emotions), social processes (e.g., maintenance of interpersonal relationships), as well as physiological processes (e.g., reduce/enhance arousal). Schäfer and colleagues conducted an empirical investigation of 129 non-redundant functions, suggesting three distinct dimensions: achieving self-awareness, regulating mood and arousal, and expressing social relatedness.

To measure interindividual variability in music use, several self-report questionnaires have been developed, most of which broadly probe the above-mentioned basic dimensions of musical functions. One key example is the 30-item scale that we utilized in the present context: the Music use Motivation (MM) module of the Music Use and Background Questionnaire (MUSEBAQ) [25], which comprises the four dimensions of musical transcendence, emotion regulation, social bonding, and musical identity and expression. The *transcendence* subscale contains items that tap into eudaimonic, cognitive, and creative functions, in addition to strong sensations and transcendental experiences. The *emotion regulation* subscale captures the tendency to use music-related strategies to regulate or alleviate (negative) emotions, for example, through diversion, discharge, or solace. The *social bonding* subscale probes motivations related to social-relational regulatory strategies, and lastly, items of the *identity* subscale relate to the expression of identity through music.

Another relevant self-report scale is the Uses of Music Inventory (UMI) [15], which contains 15 items categorizing music use into the three main factors of emotional regulation (e.g., to induce, alter, or maintain emotional states), background use (e.g., listening to music while working), and cognitive use (e.g., using music for intellectual stimulation). Although there is some overlap among the UMI and MM subscales (e.g., both include an 'emotion regulation' dimension), a notable strength of the MM is that it expands the phenotypic spectrum to include dimensions like musical transcendence, which occupies a central position in some theoretical models of musical functions [e.g., 28] and aligns with qualitative accounts describing strong experiences with music [29].

But why do people differ in their preferred ways of engaging with music? Investigating the origins and nature of individual differences in music use motivations may comprise two central questions: 1) how do genes and environments contribute to individual differences? 2) what role does personality play? A thorough examination of both questions is not only essential for developing a comprehensive model of different modes of music engagement but could also prove to be useful for future research exploring the complex relationships between music engagement, mental health, and well-being. It could furthermore contribute to a deeper understanding of 'musical sensibility,' conceptualized as the disposition to be emotionally and aesthetically engaged by or *feeling* the music [7].

## Individual differences: The role of genetic and environmental influences

To begin to answer questions about the origins of individual differences, a useful approach is to employ the classical twin design (CTD) to disentangle the relative contributions of genetic and environmental factors to trait variability. Over the past decades, a growing body of research has shown that most traits that fall within the broad definition of musicality, or the ability to perceive, produce, and enjoy music [30], are influenced by genetic effects. For example, the average heritability, i.e., the amount of trait variability that can be attributed to genetic factors [31], for objectively measured perceptual abilities (e.g., rhythm, melody, and pitch discrimination skills) and self-reported musical expertise (e.g., music aptitude or achievement) has been estimated to be around 42% [32].

Although the genetically informed literature on musicality has greatly developed in the last decades, investigations of the etiology of emotional and aesthetic musical traits have only recently emerged [7,8]. One example is a previous study reporting substantial heritability (49–64%) for self-reported 'musical sensibility' [7]. The affective response that a listener can experience when engaging with music, e.g., the degree of musical sensibility, is likely to be related to the goals and motivations for music engagement (i.e., the outcome, benefit or effect a listener is pursuing). However, there can be marked differences in the etiology of phenotypes that appear, on a surface level, closely related. Thus, our first aim is to provide a baseline understanding of the genetic and environmental bases of interindividual variability in the music use motivations.

## Individual differences: Linking music use motivations and personality traits

Personality traits can be defined as relatively stable, individual differences in cognition, affect, and behavior [33]. In general, personality traits are often referred to as 'broad' when they relate to a more general set of tendencies (e.g., agreeableness) and 'narrow' when they pertain to more specific behaviors (e.g., compassionate, trusting, respectful). Over the past decades, there has been growing interest in associations between personality, specifically the Big Five, and various musical traits, with research focusing, for example, on their links to preferences, musical sophistication, and musical sensibility [e.g., 34–36]. Most relevant for the present context, however, is the empirical work that specifically examined the relation between individual patterns of music use and the Big Five, wherein music use most often has been assessed using the UMI [13–17]. As previously mentioned, consistent relationships have been found between open-mindedness and cognitive uses of music, with studies reporting small to moderate associations (i.e., correlations and standardized betas ranging from 0.13–0.38) [13–17]. In terms of the link between negative emotionality and musical emotion regulation, a meta-analysis [37] indicated a small-to-medium summary effect ($r = 0.22$, 95% confidence interval (CI) [0.17–0.27]) based on 13 correlational studies, nine of which utilized the emotion regulation subscale of the UMI. Fewer and weaker associations have typically been reported for the other three traits of the Big Five. For instance, only one study has reported a significant negative link between conscientiousness and musical emotion regulation ($r = -0.22$, $p < 0.01$) [15], whereas none of the previous studies have found support for links between agreeableness and any of the music use subscales. Results have been considerably more equivocal regarding extraversion, with studies reporting small (e.g., $|r| < 0.20$) positive, negative, or non-significant associations with emotional and background use of music [13,15–17].

While providing important initial insights, this previous work has key limitations, particularly concerning the psychometric and conceptual properties of the UMI, since there is little available data on its reliability and validity. Crucially, its limited conceptual bandwidth prevents a more comprehensive understanding of the associations between music use and personality traits. Finally, few of the prior studies have examined the facet-level associations with the Big Five and music use [but see, e.g., [38]. This may have masked more nuanced perspectives, as facets typically provide a richer and more precise view of criteria of interest, and there is robust evidence showing that they demonstrate modest but meaningful incremental predictive validity compared to broad traits [33,39,40], especially in terms narrow outcomes [41].

Besides these methodological shortcomings, there are also central questions that remain unanswered. The first concerns the associations with Trait Empathy – broadly defined as the dispositional tendency to understand and feel the experiences of others and which is typically considered to comprise both cognitive and emotional components [42]. The cognitive component reflects the tendency to consciously detect or understand how others think and feel, whereas the emotional component reflects the tendency to experience the perceived or assumed affective state of others. The four-dimensional Interpersonal Reactivity Index (IRI) [42,43] is a widely used self-report measure of dispositional empathy and covers the facets of fantasy, perspective taking, empathic concern, and personal distress, which tap into fictionally evoked empathy, mentalizing processes, compassion, and emotion sharing, respectively. Combined, fantasy and perspective taking are assumed to index cognitive empathy, whereas empathic concern and personal distress represent the affective aspects of empathy.

Although there is no direct empirical evidence for associations between empathy and general music use traits, global empathy has, for, example, been shown to be a robust predictor ($\beta = 0.40$, $p < 0.001$) of emotional responding to music [44], and the fantasy facet in particular is associated with different facets of music reward sensitivity [18]. Moreover, two empathy facets, empathic concern and fantasy, have repeatedly been linked to the paradoxical enjoyment of nominally sad music [19,20,45,46] and the closely associated phenomenon of experiencing profound feelings of being moved or touched by music [21,46]. Indeed, according to the 'Pleasurable compassion theory' proposed by Huron and Vuoskoski [47], individuals scoring high on empathic concern and fantasy but moderate on the personal distress facet may be motivated to engage with music because they more readily experience the pleasurable and prosocial feeling of music-induced compassion, which may be experienced as intrinsically rewarding both within and outside the musical domain.

As noted, there is considerable overlap among the Big Five and empathy facets. For example, using the IRI and two measures of the Big Five, Mooradian and Davis [22] found that empathic concern, personal distress, and fantasy were positively associated with agreeableness, negative emotionality, and open-mindedness, respectively; whereas complex and interstitial associations to the Big Five were found for perspective taking. Overall, the Big Five measures explained the greatest amount of variance in the empathic concern facet ($R^2 = 0.36$–$0.44$) but the least in the fantasy facet ($R^2 = 0.12$–$0.19$), suggesting that fantasy facet lies outside the Big Five space.

The second unanswered question relates to the nature of the relationship between music use and personality. Specifically, one possible explanation that has not yet been explored is that they may be associated because of shared genetic and/or environmental influences. Indeed, it is well-established that personality is moderately heritable, with genetic effects typically accounting for 40–50% of the variance [48–51]. Moreover, research on the relations between musical traits and other non-musical psychological characteristics (e.g., personality, intelligence) points to shared genetic influences as the main etiological factor in these associations [52–55]. Extending this line of inquiry, we recently found, using the CTD, considerable phenotypic and predominantly genetic overlap between global musical sensibility and open-mindedness (especially the aesthetic sensitivity facet), and, to a lesser extent, agreeableness (especially the compassion facet) and negative emotionality [36]. Thus, one possible account is that music use motivations and personality are linked due to common genetic and/or environmental influences.

In sum, although the existing research has succeeded in identifying key phenotypic associations between selected dimensions of music use and the Big Five, more research is needed regarding other aspects of personality and the role of shared genetic and environmental factors. Relying on well-validated and comprehensive assessments of music use motivations and personality, combining different but key personality frameworks, our second aim is to provide a detailed understanding of how music use motivations relate to different aspects of personality, both phenotypically and etiologically. Given that most previous studies have focused on domain-level associations and the previously noted advantages of using conceptually narrower personality traits, we focused here on higher-resolution associations with personality facets.

## The present study

In seeking to address the two principal aims of this study, our primary objective was to provide a detailed understanding of the origins and nature of individual differences in music use motivations. To this end, we employed phenotypic- and twin analyses to examine a) the genetic and environmental basis of motivations for music use (aim 1), and b) the pattern of associations and degree of etiological overlap between the music use motivation dimensions and personality, including facets of the Big Five and empathy (aim 2).

## Methods

### Procedure and ethics

The data used in the present context was collected online from March 24th through April 19th 2022. The overall project was approved by the Regional Committee for Medical and Health Research Ethics of South-East Norway (#244965).

Participants were recruited from the Norwegian Twin Registry [56] through email and/or text message invitations, which included information about the study and a link to the survey webpage. Written informed consent was obtained from all participants, and all methods were performed in accordance with relevant guidelines and regulations.

## Participants

The sample for the present paper consisted of same-sex monozygotic (MZ) and dizygotic (DZ) twins born 1967–1991. In total, 2611 individual twins signed an informed consent and agreed to participate in the study (response rate = ~33%). This sample consisted of 1242 paired responders and 1369 single responders, of which 1728 were women (mean (SD) age = 43.4 (7.75), range = 31–55) and 883 were men (mean (SD) age = 44.5 (7.31), range = 31–55). Of the full twin pairs, there were 431 MZ pairs and 190 DZ pairs.

## Measures

*Music use motivations* were measured using Module 4 of the MUSEBAQ inventory [25]. As described, it consists of four subscales, including musical transcendence (MM-transcendence; 10 items), emotion regulation (MM-emotion regulation; 9 items), social bonding (MM-social; 7 items), and music identity and expression (MM-identity; 4 items). An example item is ''*Music raises me to another state of mind.'* Supplementary S1 Table in S1 File gives an overview of all items and descriptive statistics. Items are rated on a 5-point scale ranging from *strongly disagree* to *strongly agree* and introduced with the prompt: '*Please tell us about your reasons for using music by ticking the circle that describes you best*.' In the current sample, Cronbach alphas ranged from 0.78 (MM-identity) to 0.93 (MM-emotion regulation).

The *Big Five traits* were assessed with the BFI-2 [12,57]. This 60-item inventory covers the five broad domains (extraversion, agreeableness, conscientiousness, negative emotionality, and open-mindedness) and three lower-order facets for each domain. For example, the open-mindedness domain includes the facets of intellectual curiosity, aesthetic sensitivity, and creative imagination. An example item is '*I am someone who is complex, a deep thinker.'* All items are rated on a 5-point scale ranging from *strongly disagree* to *strongly agree.* A full overview of items can be found in Soto and John [12]. In the current sample, Cronbach alphas ranged from 0.55 (the respectfulness facet of agreeableness) to 0.82 (the sociability facet of extraversion and depression of negative emotionality).

*Trait Empathy* was assessed using the Brief Interpersonal Reactivity Index (B-IRI) [58]. This shortened version of the original IRI [42] consists of 16 items assessing individual differences in empathy across the facets of fantasy, perspective taking, empathic concern, and personal distress. An example item is '*I often have tender, concerned feelings for people less fortunate than me.'* Items are rated on a 7-point scale ranging from *does not describe me at all* to *describes me very well*. A full overview of items can be found in Ingoglia et al., [58]. Cronbach alphas in the current sample ranged from 0.75 (personal distress) to 0.84 (fantasy).

## Statistical analyses

All statistical analyses were performed using *R* (versions 4.4.1 and 4.4.2) [59] and *R Studio* (versions 2024.04.2.764 and 2024.9.1.394) [60]. Phenotypic correlations were calculated based on a sample of single twins and one member from each twin pair (*n* ranged from 1961−1987) to minimize bias. In the regression analyses, Generalized Estimating Equations (GEE) [61] and the R package *geepack* (version 1.3.12) [62] were used to account for the dependence of the twin data. The R packages *umx* (versions 4.19.00 and 4.21.0) [63] and *OpenMx* (versions 2.20.6 and 2.21.13) [64] with full information maximum likelihood estimation were used for all twin analyses. Model fit was determined using the minus-2-log-likelihood test ($\Delta-2LL$) and the Akaike's Information Criterion (AIC) [65]. A non-significant test indicates that a nested model does not cause a significant decrease in fit relative to the full model and that the more parsimonious model is preferred, whereas a lower AIC value indicates a better fit.

**Preliminary analyses.** Before the main statistical analyses, we conducted initial checks on the data. Since the MM scale is unbalanced (i.e., it contains no negatively worded items), preliminary analyses included preprocessing of the MM variables to estimate and correct for the influence of acquiescence. To this end, we utilized the BFI-2, which is perfectly balanced, to compute an acquiescence score for each respondent. This score was then used to adjust the raw MM scores for acquiescence effects using regression procedures. Significant acquiescence effects were found for all four subscales and explained, on average, 4% of the composite MM scores (adjusted $R^2$ range = 0.03–0.06; S2 Table in S1 File). Standardized residual scores were used in all analyses except for descriptive statistics.

**Twin design.** The CTD includes MZ twins (who share 100% of their genetic material) and DZ twins (who share, on average, 50% of their genetic material) and capitalizes on the differences in genetic relatedness to partition the variance and covariance into genetic and environmental components. These include additive genetic influences (A), dominance genetic influences (D), shared environmental (C; common environmental influences that make individuals within a twin pair become more similar), and unique environmental influences (E; individual-specific environmental influences that make members of a twin pair become less similar, in addition to measurement error). Different patterns of within-pair (intraclass) correlations can be used to infer the relative impact of the latent variance components. As a heuristic, A effects are inferred when MZ intraclass correlations are larger than the DZ intraclass correlations; C effects are suggested if the MZ intraclass correlations are less than twice the DZ intraclass correlations; D effects are indicated if the MZ intraclass correlations are more than twice the DZ correlation. The E component is always included as it also comprises measurement errors.

**Data analytical approach.** Prior to genetic modeling, assumptions of the classical twin design were tested by constraining means and variances to be equal within pairs and across zygosity, whereby no significant differences were found (all $p$'s ≥ 0.061; S3-S5 Tables in S1 File). Next, all variables were adjusted for sex and age using the *umx_residualize* function to avoid potential parameter biases [66]. Intraclass correlations were computed using the *umxSummarizeTwinData* function. We began by assessing the etiology of the MM dimensions by fitting a four-variate ACE biometric model using the direct variance component approach [67]. The full ACE model was compared with nested sub-models to examine whether the exclusion of either the A or C variance components would cause a significant loss of fit.

Next, we examined the covariance between MM and personality traits in two analytical steps. First, associations were assessed at the phenotypic level, beginning with the calculation of phenotypic correlations, followed by a set of multiple regression models to establish the independent effects of personality facets for each of the MM dimensions. Age and sex were included as covariates. Second, to examine the etiological overlap between the MM dimensions and personality facets, one multivariate biometric model was fitted for each MM dimension. To focus on empirically robust associations and reduce the complexity of the models, only personality facets with standardized beta coefficients meeting the more stringent significance threshold of $p < 0.01$ were included. We began by fitting a Cholesky model, wherein personality variables were included first (entered from the strongest to the weakest beta coefficient), and the respective MM dimension was entered last. This approach facilitates the estimation of the proportion of shared (personality-related) vs. unique genetic and environmental variance in the MM dimensions. Finally, based on the best-fitting Cholesky model, we used the direct variance approach to estimate genetic ($rA$) and environmental ($rE$) correlations and 95% CIs.

**Adjustment of significance threshold (alpha level) for multiple testing.** Note that we did not apply multiple testing correction to the correlational results, since these were primarily intended to provide a broad, descriptive overview of the patterns of associations. In the regression models, we specified a more stringent significance threshold ($p < 0.01$). Such specifications, rather than adjustments, have been argued to be more suitable when interested in screening associations between several variables to identify the largest and most reliable relationships and, presumably, those that are more likely to represent substantively meaningful effects [68,69]. Nevertheless, to ensure that the reported regression results were robust, we report both unadjusted and adjusted $p$-values.

*P*-values were corrected for multiple testing using the Benjamini & Hochberg False Discovery Rate (FDR) procedure [for an overview of the different methods, see, e.g., 70], which is often considered appropriate in exploratory or descriptive research, where tolerating some false positives is acceptable in order to guard against the risk of excessive false negatives. The correction was applied using the *p.adjust* function in R, separately for each regression model, excluding the covariates age and sex.

**Sensitivity analyses.** Two sets of sensitivity analyses were performed. First, we conducted a Confirmatory Factor Analysis (CFA) to assess whether the four MM subscales reflect distinct latent dimensions (a four-factor model) or a single underlying factor (a one-factor model) using the *lavaan:cfa()* function in the *lavaan* R package (version 0.6.19) [71]. The models were compared based on traditional fit indices, including the Comparative Fit Index (CFI), Tucker-Lewis Index (TLI), Root Mean Square Error of Approximation (RMSEA), and Standardized Root Mean Square Residual (SRMR). Second, one of the items of the aesthetic sensitivity facet of open-mindedness explicitly mentions music (i.e., item 20: *Is fascinated by art, music, or literature*). To examine whether the observed associations with MM subscales primarily were driven by this particular item, we conducted additional analyses using an aesthetic sensitivity score in which this item was excluded, as has been done in prior musicality research [34,36].

# Results

Table 1 presents the total sample size (*N*), the number of complete twin pairs across zygosity, descriptive statistics, and twin correlations for each study variable. Table 2 provides the phenotypic correlations between composite MM scores. As Table 2 depicts, all correlations among the MM dimensions were strong (*r* range = 0.62–0.80) and highly significant (*p* < 0.001). Note that the CFA sensitivity analysis indicated an overall better fit for the four-factor model (CFI = 0.988, TLI = 0.987, RMSEA = 0.092, SRMR = 0.057) compared to a single-factor model (CFI = 0.983, TLI = 0.982, RMSEA = 0.108, SRMR = 0.071). In addition, none of the CIs around the phenotypic correlations included 1 (Table 2). Together, these findings provide initial evidence that the subscales capture distinct, though related, dimensions.

## Genetic and environmental influences on motivations for music use

The pattern of MZ and DZ twin correlations indicated evidence of additive genetic influences for all dimensions (Table 1). Potential C effects were indicated (i.e., *r*MZ < 2 x *r*DZ) for the MM-social subscale, whereas the DZ correlations were about half that of the MZ twin correlations for the other MM subscales. Model comparisons showed that the AE model had the best fit to the data (ΔAIC = −8.912; *p* = 0.351; S6 Table in S1 File). The heritability estimates ranged from 0.38 (MM-Social) to 0.52 (MM-Identity), all of which were significant based on the 95% CIs (Table 3). In the full ACE model, however, C effects were found for the MM-Social subscale (estimated at 0.29, 95% CI [0.01; 0.53]). We therefore fitted a univariate model specifically for this subscale to determine the significance of such effects. Neither the AE model (*p* = 0.052) nor the CE model (*p* = 0.613) fit significantly worse than the full ACE model, but the CE model had the best fit according to the AIC (ΔAIC −1.74). In this model, the shared environmental variance component was estimated at 0.34, 95% CI [0.27; 0.41]. However, given the general lack of C effects previously reported for conceptually related measures [7,8] and the improved AIC results when all C components were collectively dropped from the multivariate model, we chose to proceed with the AE model in subsequent analyses.

As depicted in Table 3, genetic correlations between the four subscales were all significant and substantial, ranging from 0.71, 95% CI [0.63; 0.78] to 0.90, 95% CI [0.86; 0.93] (mean *r*A = 0.80). The unique environmental correlations were also high, ranging from 0.48, 95% CI [0.42; 0.55] to 0.75, 95% CI [0.71; 0.79] (mean *r*E = 0.60). Of note, the phenotypic- (*r* = 0.80), genetic- (*r*A = 0.90), and environmental (*r*E = 0.69) correlations between the MM-transcendental and MM-identity subscales were all substantial, indicating a high degree of shared variance among these two specific subscales.

**Table 1. Sample description, descriptive statistics, and twin correlations for all study variables.**

| Music use motivations | Descriptives | | | | | | | Twin correlations | |
|---|---|---|---|---|---|---|---|---|---|
| | Total N | MZ/DZ pairs | Range | α | Mean (SD) | Skew | Kurtosis | rMZ (SE) | rDZ (SE) |
| MM-Transendence | 2580 | 418/188 | 1-5 | 0.92 | 2.83 (0.85) | −0.11 | −0.37 | 0.50 (0.04) | 0.22 (0.07) |
| MM-Emotion regulation | 2579 | 418/188 | 1-5 | 0.93 | 2.62 (0.91) | 0.12 | −0.59 | 0.42 (0.04) | 0.15 (0.07) |
| MM-Social | 2580 | 418/188 | 1-5 | 0.81 | 2.80 (0.70) | −0.27 | −0.09 | 0.36 (0.04) | 0.31 (0.07) |
| MM-Identity | 2582 | 418/188 | 1-5 | 0.78 | 2.93 (0.88) | −0.07 | −0.46 | 0.52 (0.04) | 0.21 (0.07) |
| **Personality facets** | | | | | | | | | |
| *BFI Extraversion* | | | | | | | | | |
| E1 Sociability | 2605 | 426/190 | 1-5 | 0.82 | 3.35 (0.94) | −0.23 | −0.57 | 0.53 (0.03) | 0.15 (0.07) |
| E2 Assertiveness | 2604 | 425/190 | 1-5 | 0.74 | 3.15 (0.82) | −0.15 | −0.55 | 0.46 (0.04) | 0.07 (0.07) |
| E3 Energy level | 2605 | 426/190 | 1-5 | 0.69 | 3.68 (0.78) | −0.39 | −0.27 | 0.45 (0.04) | 0.21 (0.07) |
| *BFI Agreeableness* | | | | | | | | | |
| A1 Compassion | 2605 | 426/190 | 1-5 | 0.66 | 4.30 (0.62) | −1.03 | 0.96 | 0.36 (0.04) | 0.14 (0.07) |
| A2 Respectfulness | 2605 | 426/190 | 1-5 | 0.55 | 4.06 (0.60) | −0.36 | −0.47 | 0.41 (0.04) | 0.25 (0.07) |
| A3 Trust | 2605 | 426/190 | 1-5 | 0.67 | 3.66 (0.74) | −0.33 | −0.24 | 0.43 (0.04) | 0.18 (0.07) |
| *BFI Conscientiousness* | | | | | | | | | |
| C1 Organization | 2605 | 426/190 | 1-5 | 0.81 | 3.99 (0.84) | −0.77 | 0.07 | 0.35 (0.04) | 0.31 (0.07) |
| C2 Productiveness | 2605 | 426/190 | 1-5 | 0.72 | 3.99 (0.74) | −0.61 | 0.09 | 0.5 (0.04) | 0.19 (0.07) |
| C3 Responsibility | 2605 | 426/190 | 1-5 | 0.59 | 4.20 (0.61) | −0.56 | −0.21 | 0.35 (0.04) | 0.11 (0.07) |
| *BFI Negative emotionality* | | | | | | | | | |
| N1 Anxiety | 2605 | 426/190 | 1-5 | 0.81 | 2.70 (1.01) | 0.19 | −0.76 | 0.44 (0.04) | 0.26 (0.07) |
| N2 Depression | 2605 | 426/190 | 1-5 | 0.82 | 2.22 (0.90) | 0.79 | −0.03 | 0.51 (0.04) | 0.14 (0.07) |
| N3 Emotional volatility | 2605 | 426/190 | 1-5 | 0.78 | 2.61 (0.94) | 0.25 | −0.68 | 0.33 (0.04) | 0.14 (0.07) |
| *BFI Open-mindedness* | | | | | | | | | |
| O1 Intellectual curiosity | 2604 | 425/190 | 1-5 | 0.62 | 3.52 (0.78) | −0.17 | −0.35 | 0.54 (0.03) | 0.19 (0.07) |
| O2 Aesthetic sensitivity | 2605 | 426/190 | 1-5 | 0.80 | 3.22 (1.04) | −0.23 | −0.69 | 0.59 (0.03) | 0.25 (0.07) |
| O3 Creative imagination | 2604 | 425/190 | 1-5 | 0.72 | 3.62 (0.79) | −0.38 | −0.24 | 0.54 (0.03) | 0.18 (0.07) |
| *IRI Empathy* | | | | | | | | | |
| IRI Fantasy | 2601 | 426/190 | 1-7 | 0.84 | 3.88 (1.49) | 0.02 | −0.76 | 0.48 (0.04) | 0.22 (0.07) |
| IRI Perspective taking | 2601 | 426/190 | 1-7 | 0.80 | 4.76 (1.11) | −0.25 | −0.11 | 0.23 (0.05) | 0.11 (0.07) |
| IRI Empathic concern | 2601 | 426/190 | 1-7 | 0.76 | 5.56 (0.92) | −0.58 | 0.21 | 0.35 (0.04) | 0.12 (0.07) |
| IRI Personal distress | 2601 | 426/190 | 1-7 | 0.75 | 3.06 (1.20) | 0.4 | −0.28 | 0.27 (0.04) | 0.26 (0.07) |

E1, E2, etc. refer to Extraversion facet number 1, 2, etc. *Abbreviations:* α Cronbach´s alpha; *BFI* Big Five Inventory; *DZ* Dizygotic; *IRI* Interpersonal Reactivity Index; *MZ* Monozygotic; *SD* Standard Deviation.

## Phenotypic associations between music use motivations and personality

Phenotypic correlations between the MM dimensions and personality facets are illustrated in Fig 1. For reference, S7 Table in S1 File gives an overview of correlations between the MM subscales and the broad personality domains (i.e., extraversion, agreeableness, conscientiousness, negative emotionality, open-mindedness, and empathy) and S8 Table in S1 File provides an overview of correlations among all primary study variables. The four MM dimensions were associated to varying degrees with the personality facets, but most correlations were positive, significant, and of small-to-medium effect size (Fig 1). The strongest correlations across all subscales were with facets related to open-mindedness followed by empathy, in particular the facets O2-aesthetic sensitivity (*r* range = 0.33–0.53) and IRI-fantasy (*r* range = 0.28–0.41). Conversely, correlations with facets of extraversion- and conscientiousness were generally low, and some were

**Table 2. Phenotypic correlations (with confidence intervals in brackets) between the Music Use Motivation variables.**

| | Phenotypic correlations | | | |
|---|---|---|---|---|
| | MM-T | MM-E | MM-S | MM-I |
| MM-T | 1.00 | | | |
| MM-E | 0.78*** [0.76; 0.80] | 1.00 | | |
| MM-S | 0.68*** [0.66; 0.71] | 0.59*** [0.56; 0.62] | 1.00 | |
| MM-I | 0.80*** [0.78; 0.81] | 0.64*** [0.61; 0.67] | 0.62*** [0.59; 0.65] | 1.00 |

Significance codes: *$p<0.05$; **$p<0.01$; ***$p<0.001$. *Abbreviations: MM-E Emotion Regulation; MM-I Musical Identity and Expression; MM-S Social; MM-T Musical Transcendence.*

**Table 3. Estimated additive genetic- (A) and non-shared environmental (E) influences, as well as genetic (below diagonal) and environmental (above diagonal) correlations.**

| Variable | Standardized variance components | | Genetic and environmental correlations | | | |
|---|---|---|---|---|---|---|
| | A [95% CI] | E [95% CI] | MM-T [95% CI] | MM-E [95% CI] | MM-S [95% CI] | MM-I [95% CI] |
| MM-T | 0.50 [0.43; 0.56] | 0.50 [0.44; 0.57] | – | 0.75 [0.71; 0.79] | 0.59 [0.53; 0.64] | 0.69 [0.64; 0.73] |
| MM-E | 0.40 [0.32; 0.47] | 0.60 [0.53; 0.68] | 0.85 [0.80; 0.89] | – | 0.48 [0.42; 0.55] | 0.54 [0.47; 0.60] |
| MM-S | 0.38 [0.30; 0.45] | 0.62 [0.55; 0.70] | 0.80 [0.73; 0.86] | 0.75 [0.66; 0.84] | – | 0.55 [0.49; 0.61] |
| MM-I | 0.52 [0.46; 0.58] | 0.48 [0.42; 0.54] | 0.90 [0.86; 0.93] | 0.80 [0.73; 0.86] | 0.71 [0.63; 0.78] | – |

*Abbreviations: A Additive genetic influences; CI Confidence Interval; E Non-shared environmental influences; MM-E Emotion Regulation; MM-I Musical Identity and Expression; MM-S Social; MM-T Musical Transcendence.*

insignificant. Further, and as expected, substantial correlations were also found among and across the Big Five and empathy variables (S8 Table in S1 File).

To identify personality traits with independent effects and to inform the selection of variables for further twin modeling, we fitted four regression models (Table 4) in which all Big Five and empathy facets served as predictors of each of the four MM subscales. Overall, personality explained 37% (MM-transcendence), 28% (MM-emotion regulation), 18% (MM-social), and 29% (MM-identity) of the phenotypic variance. Of the 19 facets, seven were significant at $p<0.01$ for MM-transcendence; six for MM-emotion regulation; five for MM-social; and three for MM-identity. Particularly the personality facets of O2-aesthetic sensitivity and IRI-fantasy represented consistent and robust predictors across all four MM dimensions. C3-responsibility and A3-trust were also significant predictors across all MM dimensions except MM-identity, but their influence was comparatively weak (i.e., beta's$<|0.10|$). Notably, the sensitivity analysis using an O2-aesthetic sensitivity score without the item that explicitly mentions music (i.e., item 20) showed that the associations with all MM dimensions remained substantial (MM-T $r=0.48$, $\beta=0.32$; MM-E $r=0.30$, $\beta=0.20$; MM-S $r=0.34$, $\beta=0.26$; MM-I $r=0.43$, $\beta=0.32$), all highly significant ($p<0.001$). Moreover, the O2-aesthetic sensitivity facet remained the strongest predictor across all MM-dimensions.

In addition to these broad patterns, specific facets had significantly unique contributions according to the type of MM. That is, MM-transcendence was further positively predicted by traits such as O1-intellectual curiosity and IRI-perspective taking, while for MM-emotion regulation, positive contributions were also found from the negative emotionality facets N2-depression and N1-anxiety. Additional positive predictors of MM-social and MM-identity included, respectively, A1-compassion and E3-energy level. In contrast to IRI-fantasy, the effects of the other empathy facets (particularly empathic concern and personal distress) were largely insignificant when accounting for the Big Five, despite non-trivial and significant correlations with certain MM dimensions.

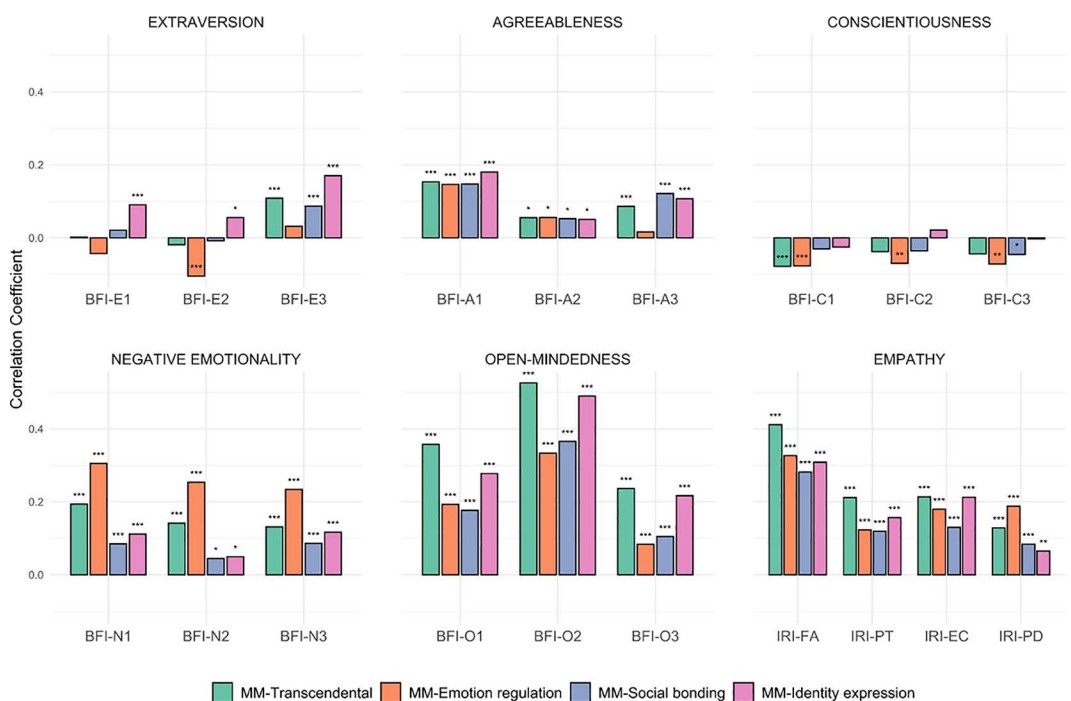

**Fig 1. Bivariate phenotypic correlations between the music use motivation subscales and the Big Five and empathy facets.** Significance codes (without adjusting for multiple testing): * p < 0.05; ** p < 0.01; *** p < 0.001. *Facet codes (in the order they appear in the plot): BFI-E1* sociability (extraversion); *BFI-E2* assertiveness (extraversion); *BFI-E3* energy level (extraversion); *BFI-A1* compassion (agreeableness); *BFI-A2* respectfulness (agreeableness); *BFI-A3* trust (agreeableness); *BFI-C1* organization (conscientiousness); *BFI-C2* productiveness (conscientiousness); *BFI-C3* responsibility (conscientiousness); *BFI-N1* anxiety (negative emotionality); *BFI-N2* depression (negative emotionality); *BFI-N3* emotional volatility (negative emotionality); *BFI-O1* intellectual curiosity (open-mindedness); *BFI-O2* aesthetic sensitivity (open-mindedness); *BFI-O3* creative imagination (open-mindedness); *IRI-FA* fantasy (empathy); *IRI-PT* perspective taking (empathy); *IRI-EC* empathic concern (empathy); *IRI-PT* personal distress (empathy).

All predictors that met our original $p < 0.01$ threshold remained significant at $p < 0.05$ following FDR correction (Table 4), and as such, supports the robustness of our findings. Moreover, the specification approach also screened out other, potentially less important, predictors that were originally significant at 0.05 and which remained significant at 0.05 after corrections (e.g., the IRI-personal distress facet as a predictor of the MM-transcendence subscale). That said, it is generally not recommended to adjust alpha levels to account for multiple testing once more stringent significance thresholds have been specified [68,69].

## Etiology of associations between personality facets and music use motivations

We first modeled the relationship between each of the MM dimensions and personality facets (with beta coefficients meeting the $p < 0.01$ significance threshold) using the Cholesky decomposition approach. Goodness-of-fit measures and results from the model comparisons summarized in S9 Table in S1 File showed that the AE model had the best fit in each case, including the MM-social subscale. An overview of the standardized parameter estimates derived from the best-fitting AE Cholesky models and the genetic and environmental correlations among all modeled variables (estimated using the direct variance approach) can be found in S10-S13 Tables in S1 File. Of note, significant genetic influences were found for all personality facets, with heritability estimates ranging from 0.24, 95% CI [0.16; 0.32] to 0.58, 95% CI [0.52; 0.63] (S14 Table in S1 File).

**Table 4. Results of regression models for the associations between personality facets and music use motivations.**

| | MM-T | | | MM-E | | | MM-S | | | MM-I | | |
|---|---|---|---|---|---|---|---|---|---|---|---|---|
| | β | Unadj. *p* | Adj. *p* | β | Unadj. *p* | Adj. *p* | β | Unadj. *p* | Adj. *p* | β | Unadj. *p* | Adj. *p* |
| Age | | | | −0.10 | < 0.001 | | | | | −0.04 | 0.033 | |
| Sex | | | | 0.09 | < 0.001 | | | | | −0.05 | 0.017 | |
| *BFI Extraversion* | | | | | | | | | | | | |
| E1 Sociability | | | | | | | | | | | | |
| E2 Assertiveness | | | | | | | | | | | | |
| E3 Energy level | 0.05 | 0.04316 | 0.082 | | | | | | | **0.08** | 0.0010 | 0.0061 |
| *BFI Agreeableness* | | | | | | | | | | | | |
| A1 Compassion | | | | | | | **0.07** | 0.0045 | 0.0171 | | | |
| A2 Respectfulness | | | | | | | | | | | | |
| A3 Trust | **0.07** | < 0.001 | 0.004 | **0.07** | 0.0029 | 0.0092 | **0.09** | < 0.001 | 0.0018 | 0.05 | 0.0292 | 0.0948 |
| *BFI Conscientiousness* | | | | | | | | | | | | |
| C1 Organization | | | | | | | | | | | | |
| C2 Productiveness | | | | 0.06 | 0.0153 | 0.0415 | | | | | | |
| C3 Responsibility | **−0.07** | 0.00213 | 0.0067 | **−0.09** | < 0.001 | 0.0010 | **−0.09** | < 0.001 | 0.0018 | −0.05 | 0.0248 | 0.0948 |
| *BFI Negative emotionality* | | | | | | | | | | | | |
| N1 Anxiety | 0.05 | 0.04177 | 0.082 | **0.10** | < 0.001 | 0.0024 | | | | | | |
| N2 Depression | **0.12** | < 0.001 | < 0.001 | **0.18** | < 0.001 | < 0.001 | | | | 0.06 | 0.0299 | 0.0948 |
| N3 Emotional volatility | | | | | | | | | | 0.05 | 0.0436 | 0.1036 |
| *BFI Open-mindedness* | | | | | | | | | | | | |
| O1 Intellectual curiosity | **0.07** | 0.00135 | 0.0051 | | | | | | | | | |
| O2 Aesthetic sensitivity | **0.38** | < 0.001 | < 0.001 | **0.24** | < 0.001 | < 0.001 | **0.30** | < 0.001 | < 0.001 | **0.42** | < 0.001 | < 0.001 |
| O3 Creative imagination | | | | | | | | | | | | |
| *IRI Empathy* | | | | | | | | | | | | |
| IRI Fantasy | **0.18** | < 0.001 | < 0.001 | **0.13** | < 0.001 | < 0.001 | **0.14** | < 0.001 | < 0.001 | **0.09** | < 0.001 | < 0.001 |
| IRI Perspective taking | **0.06** | 0.00566 | 0.0154 | | | | | | | | | |
| IRI Empathic concern | | | | | | | | | | 0.05 | 0.0399 | 0.1036 |
| IRI Personal distress | 0.05 | 0.0162 | 0.0385 | | | | 0.05 | 0.027 | 0.0862 | | | |
| Marginal R² | 0.37 | | | 0.28 | | | 0.18 | | | 0.29 | | |

Regression analyses were estimated using generalized estimating equations. Coefficients in bold are statistically significant at our pre-specified threshold *p* < 0.01 (unadjusted p-value). *Abbreviations: Adj. p* FDR-adjusted p-value; β standardized beta coefficient; *BFI* Big Five Inventory; *IRI* Interpersonal Reactivity Index; *MM-E* Emotion Regulation; *MM-I* Music Identity and Expression; *MM-S* Social; *MM-T* Music Transcendence; *Unadj. p* unadjusted p-value.

Fig 2 depicts the genetic and environmental variance decomposition of each MM dimension, distinguishing between personality-related influences and those unique to each dimension. Fig 3 displays the pairwise genetic and unique environmental correlations (and their 95% CIs) between the MM dimensions and personality facets included in the models.

Seven facets were included in the MM-transcendence twin models: O2-aesthetic sensitivity, IRI-fantasy, N2-depression, A3-trust, C3-responsibility, O1-intellectual curiosity, and IRI-perspective taking. Variance decomposition showed that personality facets explained 33% and 9% of the genetic and environmental variance, respectively (Fig 2A). Of the total genetic variance in MM-transcendence (50%), the majority was explained by personality-related genetic influences (66%), whereas personality explained 17% of the total environmental variance (50%). The genetic correlations with the open-mindedness- and empathy facets were all substantial (Fig 3), ranging from 0.51, 95% CI [0.42; 0.60] (O1-intellectual curiosity) to 0.72, 95% CI [0.65; 0.79] (O2-aesthetic sensitivity). Except for IRI-perspective taking, all

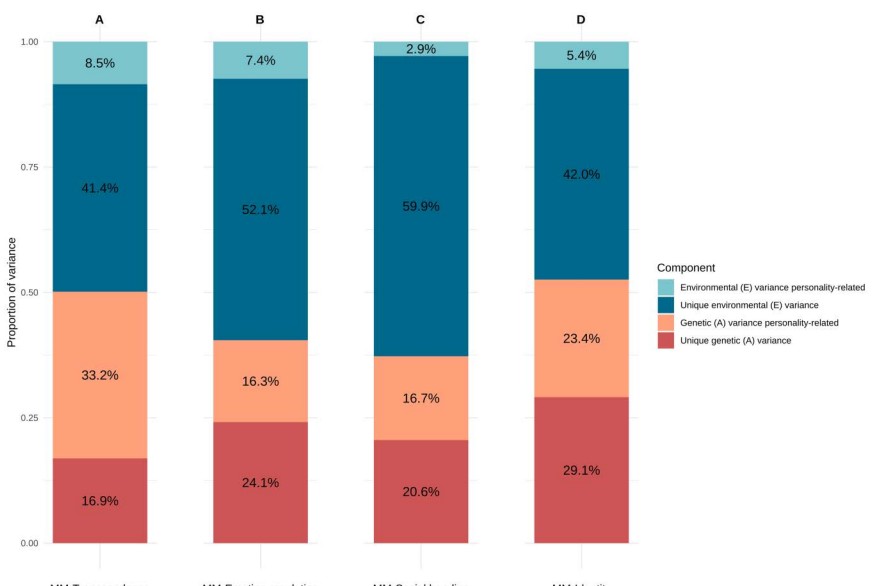

**Fig 2. Proportion of variance of the MM-transcendence (A), MM-emotion regulation (B), MM-social bonding (C), and MM-identity (D) subscales explained by genetic and environmental factors specific to each of them (unique genetic/environmental) or overlapping genetic/environmental factors with personality.**

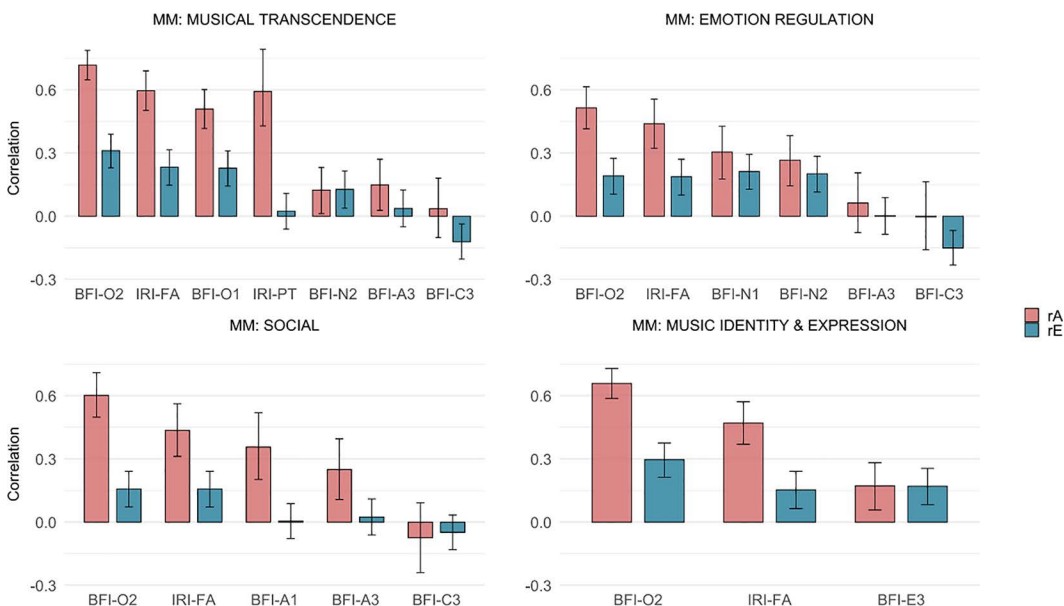

**Fig 3. Estimates of genetic (*r*A) and non-shared environmental (*r*E) correlations and their 95% confidence intervals between the MM dimensions and the personality variables.** *Notes.* Personality variables are sorted based on the magnitude of the phenotypic correlations (cf. Fig 1). *Facet codes: BFI-A1* compassion (agreeableness); *BFI-A3* trust (agreeableness); *BFI-C3* responsibility (conscientiousness); *BFI-E3* energy level (extraversion); *IRI-FA* fantasy (empathy); *BFI-N1* anxiety (negative emotionality); *BFI-N2* depression (negative emotionality); *BFI-O1* intellectual curiosity (open-mindedness); *BFI-O2* aesthetic sensitivity (open-mindedness); *IRI-PT* perspective taking (empathy).

environmental correlations were significant and small to moderate, ranging from 0.23 for both O1-intellectual curiosity (95% CI [0.14; 0.31]) and IRI-fantasy (95% CI [0.15; 0.32]) to 0.31, 95% CI [0.23; 0.39] (O2-aesthetic sensitivity). Estimates of genetic and environmental correlations with the other facets included in the model, i.e., N2-depression, A3-trust, and C3-responsibility, were generally small (< 0.15) with large confidence intervals.

Six facets were included in the MM-emotion regulation twin models: O2-aesthetic sensitivity, N2-depression, IRI-fantasy, N1-anxiety, C3-responsibility, and A3-trust. Variance decomposition showed that personality facets explained 16% and 7%, respectively, of the genetic and environmental variance in MM-emotion regulation (Fig 2B). Of the total genetic (40%) and environmental (60%) variance, personality-related influences explained 40% and 12%, respectively. As depicted in Fig 3, positive and moderately high genetic correlations were found between MM-emotion regulation and the personality facets O2-aesthetic sensitivity ($r_A = 0.51$, 95% CI [0.41; 0.61]), IRI-fantasy ($r_A = 0.44$, 95% CI [0.32; 0.56]), N1-anxiety ($r_A = 0.30$, 95% CI [0.18; 0.43]), and N2-depression ($r_A = 0.27$, 95% CI [0.14; 0.38]), whereas environmental correlations were small and similar across these facets ($r_E \approx 0.20$). Both the genetic and environmental associations with A3-trust, as well as the genetic correlation with C3-responsibility, were non-significant, whereas the environmental association with C3-responsibility was significant and weakly negative ($r_E = -0.15$, 95% CI [-0.23; -0.07]).

Five facets were included in the MM-social twin models: O2-aesthetic sensitivity, IRI-fantasy, C3-responsibility, A3-trust, and A1-compassion. Variance decomposition showed that personality facets explained 17% and 3%, respectively, of the genetic and environmental variance in MM-social (Fig 2C). Of the total genetic (38%) and environmental (62%) variance, personality-related influences explained 45% and 5%, respectively. The genetic correlations with O2-aesthetic sensitivity ($r_A = 0.60$, 95% CI [0.50; 0.71]) and IRI-fantasy ($r_A = 0.44$, 95% CI [0.31; 0.56]) were moderate to strong (Fig 3), in addition to small environmental correlations ($r_E = 0.16$, 95% CI [0.07; 0.24] in both cases). The genetic correlations with the two agreeableness facets A1-compassion ($r_A = 0.36$, 95% CI [0.20; 0.52]) and A3-trust ($r_A = 0.25$, 95% CI [0.11; 0.40]) were small to moderate, whereas the environmental correlations were both non-significant. Neither correlation with C3-responsibility was significant.

Three facets were included in the MM-identity twin models: O2-aesthetic sensitivity, IRI-fantasy, and E3-energy level. Variance decomposition showed that personality facets explained 23% and 5%, respectively, of the genetic and environmental variance (Fig 2D). Of the total genetic (52%) and environmental (48%) variance, personality-related influences explained 44% and 10%, respectively. MM-identity had strong genetic correlations and small to moderate environmental correlations with O2-aesthetic sensitivity ($r_A = 0.66$, 95% CI [0.59; 0.73], $r_E = 0.30$, 95% CI [0.21; 0.38]) and IRI-fantasy ($r_A = 0.47$, 95% CI [0.37; 0.57]; $r_E = 0.15$, 95% CI [0.06; 0.24]), whereas the genetic and environmental associations with E3-energy level were small and equal ($r_A = 0.17$, 95% CI [0.06; 0.28]; $r_E = 0.17$, 95% CI [0.08; 0.25]) (Fig 3).

Although associations between MM dimensions and certain personality facets seemed to reflect trait-congruent associations, strong and consistent genetic links were found across all MM dimensions for O2-aesthetic sensitivity ($r_A$ range = 0.51–0.72, mean = 0.62) and IRI-fantasy ($r_A$ range = 0.44–0.60, mean = 0.49), in addition to small-to-moderate unique environmental correlations (O2-aesthetic sensitivity $r_E$ range = 0.16–0.31, mean = 0.24; IRI-fantasy $r_E$ range = 0.15–0.23, mean = 0.18). The two facets also shared substantial etiological variance with each other ($r_A$ and $r_E$ about 0.57, 95% CI [0.48; 0.65] and 0.19, 95% CI [0.10; 0.27], respectively; see S10-S13 Tables in S1 File). Given the complexity of the etiological relationships between personality variables, a thorough examination of their unique contribution to the genetic and environmental covariance with MM dimensions is beyond the scope of this study. However, as can be seen in S10-S13 Tables in S1 File, IRI-fantasy had significant unique genetic covariance, over and above O2-aesthetic sensitivity, for two of the four MM dimensions (MM-transcendence and MM-emotion regulation), in addition to unique environmental covariance with all MM dimensions. Importantly, these findings suggest that the associations between IRI-fantasy and MM dimensions are not solely explained by its shared variance with O2-aesthetic sensitivity, mirroring results from the regression analyses.

## Discussion

The present study sought to provide a comprehensive understanding of individual differences in key dimensions of motivations for music use, measured using the MM scale. This was addressed along two paths: First, we examined the extent to which genetic and environmental influences contribute to individual differences in the four MM dimensions, and second, we assessed their phenotypic- and etiological overlap with the Big Five and empathy traits.

### Etiology of different dimensions of music use motivation

Our first aim was to determine the relative importance of genetic and environmental factors in shaping individual differences. Heritability estimates were 50%, 40%, 38%, and 52% for MM-transcendence, MM-emotion regulation, MM-social, and MM-identity, respectively, with an average heritability of 45%. Individual-specific environmental influences accounted for the remaining variance. These estimates are consistent with the broader genetically informed literature on musicality [32], as well as the heritability of the five music reward facets (range = 42%-52%, mean = 47%) [8], but they appear to be somewhat lower than what we previously found for the four musical sensibility facets (range = 49%−65%, mean = 56%) and general musical sensibility (64%) [7]. Thus, these initial findings suggest that compared to musical sensibility, individual differences in the various dimensions of MM are more influenced by environmentally shaped unique experiences than by genetic factors.

Although the multivariate modeling results indicated significant additive genetic and non-shared environmental influences, it is worth noting that, in the univariate model for the MM-social subscale specifically, shared environment influences were significant, whereas additive genetic influences were not. This latter finding contrasts with the earlier research on the etiology of musical sensibility [7] and music reward sensitivity [8]. In fact, in the latter study, 'Social reward' had the highest heritability (52%) among the five music reward facets and the authors reported no indications of shared environmental influences.

In the present results, the lack of significant genetic effects (in the univariate model) likely reflects limited power to distinguish between familial effects, especially when one of the components (A or C) is small. Moreover, multivariate models, with more information, generally involve greater power and, consequently, more robust estimates than univariate models. Nevertheless, our results do indicate that the heritability of this subscale is likely modest and, along with its genetic correlations with other variables, should thus be interpreted cautiously. This further suggests that the genetic and environmental architecture may differ across the four subscales and that there might be something special about the MM-social subscale, which should be more comprehensively investigated in future studies using alternative modeling approaches.

Another noteworthy finding was the considerably high associations among the MM subscales: phenotypically ($r$ range = 0.59–0.80), genetically ($rA$ range = 0.71–0.90), and environmentally ($rE$ range = 0.48–0.75). Importantly, high inter-factor correlations were also reported in the original publication by Chin et al. [25] (range 0.75–0.90), suggesting that these findings are not just due to chance or specific to the present sample. These findings can be interpreted in different ways, one of which relates to questions about the distinctiveness of the four subscales. In particular, the especially strong associations between the MM-transcendence and MM-identity subscales may indicate that the two subscales reflect highly overlapping concepts. This would accord with the empirical findings by Schäfer et al. [1], wherein the first of the three identified dimensions (self-awareness) encompassed items indexing functions related to, e.g., self-related thoughts, identity, meaning, absorption, and escapism [cf. Table A3 in Schäfer et al. 1], echoing the conceptual content of the MM-transcendence and MM-identity dimensions. However, our preliminary CFA results, combined with the fact that none of the CIs around the phenotypic correlations reached 1, provide tentative support for the notion that, while related, the four subscales do not measure the same thing, supporting a multidimensional structure.

Another interpretation is that the overall high associations between dimensions reflect the presence of a common underlying latent factor, or alternative data-generating mechanisms, like those postulated by network models [72]. Since

we did not comprehensively assess the underlying latent structure, the present findings strongly warrant further examination of the structure and dimensionality of the MM scale.

## Links between music use motivations and personality

Our second aim was to examine the role of personality traits (the Big Five and Trait Empathy) in individual differences in motivations for music use. To this end, we performed a comprehensive assessment of facet-level associations across MM dimensions at the phenotypic and etiological levels.

**Phenotypic overlap between music use motivations and personality facets.** Our results indicate that personality traits predicted the MM dimensions to varying degrees, accounting for the least variance in the MM-social dimension (18%) and the most in the MM-transcendence dimension (37%). In terms of MM – personality linkages, we identified both broad and more dimension-specific patterns of associations. Regarding the former, a key finding of the present study is that among all Big Five and empathy facets, aesthetic sensitivity (open-mindedness) and fantasy (empathy) were the most prominently associated with general music use motivation, demonstrating consistent and unique effects across all MM dimensions. This is in keeping with previous evidence identifying strong associations between all facets of music reward sensitivity and fantasy [18], as well as the open-mindedness facets of 'aesthetics' and 'feelings' [73]. It also aligns with the previously reported robust association between global musical sensibility and aesthetic sensitivity [36]. Combined with a growing body of research linking open-mindedness and empathy more broadly to intense emotional and aesthetic reactions to music or other aesthetic stimuli [e.g., 19,36,46,74,75], a possible interpretation is that individuals scoring high on these traits more easily become 'absorbed' (or immersed, interested, and transported) in musical experiences; a link that has previously been shown for fantasy [76] and the 'openness' aspect of the Big Five Aspects Scale [75]. A related possibility is that a facet like fantasy taps into individual differences in psychological and brain processes that facilitate emotional engagement with music [47,77].

It is worth noting that, both in the present and in prior musicality research [34,36], sensitivity analyses showed that associations with the aesthetic sensitivity facet remained robust across MM subscales even when controlling for the BFI-2 item explicitly mentioning music. In fact, aesthetic sensitivity consistently emerged as the strongest predictor across all MM subscales, consistent with previous results. Collectively, these findings suggest that the associations with aesthetic sensitivity are unlikely to be primarily driven by content overlap with musicality-related measures.

Our findings also suggested theoretically meaningful trait profiles for most MM dimensions, providing some evidence that they captured discrete aspects of motivations for music use. For instance, in addition to the strong effects of aesthetic sensitivity and fantasy, MM-transcendence was most robustly and positively linked with the facets of intellectual curiosity (open-mindedness) and perspective taking (i.e., the other cognitive facet of empathy). Regarding open-mindedness facets, a comparable pattern was found in a study reporting correlations between the UMI and IPIP-300 [78] personality facets, wherein the intellect facet was only significantly correlated with the cognitive subscale of the UMI [38]. In contrast, the artistic interest facet showed significant correlations with all three UMI subscales. More broadly, our results also align with the well-established positive links between open-mindedness and cognitive music use [e.g., 13,15], and preference for sophisticated or complex music [e.g., 35]. These findings fit well with research showing that facets of open-mindedness typically are more strongly associated with eudaimonic rather than affective conceptualizations of well-being, specifically the personal growth component of psychological well-being (PWB) [39], broadly mirroring findings of global empathy [79].

In line with existing research, facets of negative emotionality (anxiety and depression) were most robustly linked to the MM-emotion regulation subscale. Overall, correlations with the three negative emotionality facets ($r$ range 0.23–0.31, all significant) were very similar to the meta-analytic summary estimate of the association between negative emotionality and musical emotion regulation ($r = 0.22$) [37]. Our findings further align with evidence of relations between so-called unhealthy or maladaptive music use (e.g., rumination, avoidance), which is also covered in the MM-emotion regulation subscale, and traits characterized by negative affect (depression, personal distress, negative emotionality) [5,24,80].

Concerning the MM-social subscale, trait-congruent associations were found with facets of agreeableness (trust and compassion). Although this Big Five domain has rarely been highlighted in relation to music use, the positive association between agreeableness facets and the use of music to strengthen and facilitate social relationships clearly fits well with the prosocial nature of these personality traits. This is also consistent with research indicating that agreeable people are more likely to prefer musical styles featuring themes about love and relationships [35], and to pursue (non-musical) pro-social emotion regulation goals [81]. Facets of agreeableness are also tightly and positively linked to the positive relations component of PWB [39].

Apart from the aesthetic sensitivity and fantasy facets, the only significant predictor of the MM-identity dimension was the energy level facet of extraversion. As previously noted, the existing research on musical habits and extraversion has yielded mixed results, and as will be described below, the current findings were not particularly robust, warranting cautious interpretation. However, some studies suggest that extraverts are more inclined to use music as a background to other activities [13,17], prefer nominally happy-sounding music [14] and music with rhythmic, upbeat, and electronic features [35]. Thus, it seems consistent that people scoring high on the energy level facet, which is characterized by enthusiasm and energy, are more likely to seek musical stimulation and use music to express their identity and feelings.

It should be noted that there were also some modest relationships that reached statistical significance but were less straightforward to interpret. For instance, the facets of depression (negative emotionality), responsibility (conscientiousness), and trust (agreeableness) were significant predictors of the MM-transcendence subscale. These links may partly reflect the broad conceptual scope of this specific subscale, which also includes items that capture aspects of emotion regulation (e.g., 'music is like a comforting friend to me.'). The responsibility facet of conscientiousness was also a weak negative predictor of the MM-emotion regulation subscale, which is consistent with some domain-level findings [15], but not others [e.g., 17]. Overall, however, in terms of zero-order correlations, the links between MM dimensions and facets related to conscientiousness, and to some extent, extraversion and agreeableness, were generally weak. Moreover, as will be discussed below, there are reasons to believe that at least some of these associations were spurious and should thus be interpreted cautiously. Finally, we note that aside from fantasy, the effects of other empathy facets were largely insignificant when accounting for the Big Five traits, despite non-trivial and significant zero-order correlations with certain MM dimensions. These findings echo those of Mooradian and Davis [22], who found that of the four empathy facets, the Big Five explained the least amount of variance in the fantasy facet.

**Genetic and environmental overlap between personality facets and music use motivations.** The results of our genetically informed analyses of the MM dimensions and personality associations showed that personality-related genetic and unique environmental influences accounted for, respectively, 16%−33% and 3%−9% of the variance in the MM dimensions. Thus, the emerging picture is that the phenotypic associations between MM dimensions and personality facets were largely due to their underlying shared genetic influences relative to environmental factors. Indeed, personality-related genetic influences accounted for nearly half (40% to 46%) of the total heritability in the MM-emotion regulation, MM-social, and MM-identity dimensions and an even larger proportion (66%) of the heritability in the MM-transcendence dimension. The corresponding values of the total non-shared environmental variance were comparatively lower (5% to 17%). This may mean, using the MM-transcendence dimension as an example, that a great part of the genes and, to a lesser extent, the unique environments that influence the tendencies of having an open, active, imaginative, and reflective mind also contribute to the propensity to use music for attaining meaning and transcendental experiences.

We also examined the genetic and environmental correlations, which index the extent to which the specific genetic and environmental underpinnings of one trait overlap with another, possibly reflecting pleiotropic processes. Particularly strong and consistent genetic correlations, in addition to small-to-moderate unique environmental correlations, were found across all MM dimensions for aesthetic sensitivity (mean $rA = 0.62$; mean $rE = 0.24$) and fantasy (mean $rA = 0.49$; mean $rE = 0.18$). In comparison, genetic correlations between MM dimensions and most of the other personality facets were typically only small-to-moderate (i.e., $rA < 0.30$). Overall, the present findings suggest that the genetic

and biological foundations of the aesthetic sensitivity and fantasy facets largely overlap with the genetic bases of the various MM dimensions, more so than most other personality traits in the models. The strong genetic links between the aesthetic sensitivity facet and MM domains align with our previous findings on the corresponding relationship between aesthetic sensitivity and musical sensibility [36], but also yields original evidence of etiological links between empathy and musical dispositions. Importantly, despite the high etiological overlap between aesthetic sensitivity and fantasy, fantasy had unique genetic and environmental covariances with most MM dimensions, over and above aesthetic sensitivity.

## Limitations

The work presented here should be interpreted in the context of certain limitations. First, the usual limitations of the CTD apply, including limited power to distinguish between different genetic mechanisms (i.e., additive and non-additive genetic effects) and familial influences (i.e., additive genetic and shared environmental influences) [82–84]. Indeed, the latter issue was raised for the MM-social subscale. Therefore, the present results do not rule out the presence of possible dominance genetic and shared environmental effects and the final AE models should thus be regarded as tentative. Moreover, the obtained estimates are limited to the specific sample (in terms of age, socioeconomic status, culture, and ethnicity) and the measurement that was used. The present sample mostly comprised middle-aged women from a rather culturally homogenous group. Although we controlled for sex and age effects, future studies should seek to examine sex-specific patterns and replicate the reported estimates in culturally more diverse samples and across different age groups, ideally using longitudinal designs. Indeed, previous studies points to age differences in observed music engagement styles [85], as well as associations with personality [86], in addition to cultural differences in how music is used to achieve various well-being goals [87]. For example, during the COVID-19 crisis, music was more prominently used as a resource for fostering a sense of togetherness in collectivistic cultures compared to individualistic ones [87]. Future work should also include additional measures of MM (e.g., behavioral tasks, physiological recordings) to complement self-reports, as well as alternative twin modeling approaches.

Second, it is important to highlight that genetic correlations do not imply causation, nor do they inform the directionality of the causal pathways – i.e., the genetic influence on which measure impacts the other, cannot be established in our correlational data. Third, all measures were assessed using self-reports, which might introduce shared method variance. If present, this may have artificially inflated the phenotypic- and non-shared environmental correlations but not genetic correlations. Fourth, in terms of the more modest and seemingly less congruent phenotypic associations between certain MM dimensions and facets of, e.g., conscientiousness and agreeableness, it is important to note that most of the genetic and environmental correlations underpinning these relationships were generally low and imprecise, and in some cases, both estimates were insignificant (e.g., in the link between the trust facet of agreeableness and the MM-emotion regulation dimension). As such, these relationships should be cautiously interpreted, as they may be spurious findings. Finally, while we used a comprehensive MM measure, it is likely not exhaustive. For example, it could be argued that the lack of a dimension capturing 'background' use might obscure meaningful relations to certain personality traits. Indeed, for many, the use of music may serve more utilitarian purposes, such as enhancing concentration, providing physical stimulation during exercise, offering entertainment, or creating background noise to eliminate silence. Relatedly, it should also be mentioned that there is a lack of items that capture the upregulation of positive emotions (in the emotion regulation subscale), such as using music to experience joy. That said, research suggests that negative emotions are most often the target of regulation in everyday life [88].

## Conclusions

The present research contributes to our understanding of the origins and nature of individual differences in motivations to music use in several ways. First, we showed that the four MM dimensions are moderately heritable and share

common genetic and environmental influences. Second, we rigorously examined both the phenotypic and etiological relationships between MM dimensions and personality facets, revealing both novel associations and corroborating previous results. Particularly consistent and robust patterns of associations were identified between the aesthetic sensitivity- and fantasy facets and all MM dimensions. At the same time, we also detected theoretically meaningful trait profiles, for example, that individuals who are trusting and concerned for others' well-being are more likely to use music to facilitate positive social interactions. We further showed that the phenotypic linkages between personality traits and MM dimensions were primarily attributable to correlated genes next to correlated unique environments.

Our study contributes to emerging evidence on the genetic bases of emotional and aesthetic musical dispositions. It also substantially expands the mapping of personality traits to music use constructs by including both different personality frameworks and facet-level analyses, which may prove valuable for researchers exploring the relationship between everyday modes of music engagement and well-being. The main results so far from our project strongly suggest that both musical sensibility and music use motivations are closely interwoven with personality dispositions (pertaining to affect, aesthetics, empathy, and intellect) and should thus not be viewed as fully self-contained traits. However, more research is needed to obtain a better understanding of these constructs, particularly how they are related, their content, and their associations with theoretically relevant external criteria.

## Supporting information

**S1 File.** **S1 Table. Music use motivation questionnaire.** Item and subscale descriptive statistics of the music use motivation scale. **S2 Table. Tests of acquiescence effects in the music use motivations variables. S3 Table. Tests of the assumptions of equal means and variances across twin order and zygosity in a multivariate model of the four music use motivations variables.** Models 2–4 are compared with the fully saturated model 1, showing no significant effects of constraining means and variances (all p's > .130). **S4 Table. Tests of the assumptions of equal means and variances across twin order and zygosity in a multivariate model of the four empathy facets.** Models 2–4 are compared with the fully saturated model 1, showing no significant effects of constraining means and variances (all p's > .348). **S5 Table. Tests of the assumptions of equal means and variances across twin order and zygosity in univariate models of the Big Five facets that were included in the final twin models.** Models 2–4 are compared with the fully saturated model 1, showing no significant effects of constraining means and variances (all p's > .061). **S6 Table. Model fitting results for the multivariate biometric models of the four music use motivation dimensions. S7 Table. Pairwise phenotypic correlations among music use motivation subscales and the broad personality domains. S8 Table. Pairwise phenotypic correlations among all study variables. S9 Table. Model fitting results from the multivariate biometric models of the covariance between the music use motivation subscales and personality facets. S10 Table. Model estimates derived from the best-fitting AE model of the MM-transcendence – personality associations.** Standardized genetic (A) and unique environmental (E) path estimates, as well as genetic (rA) and unique environmental (rE) correlations. **S11 Table. Model estimates derived from the best-fitting AE model of the MM-emotion regulation – personality associations.** Standardized genetic (A) and unique environmental (E) path estimates, as well as genetic (rA) and unique environmental (rE) correlations. **S12 Table. Model estimates derived from the best-fitting AE model of the MM-social – personality associations.** Standardized genetic (A) and unique environmental (E) path estimates, as well as genetic (rA) and unique environmental (rE) correlations. **S13 Table. Model estimates derived from the best-fitting AE model of the MM-identity – personality associations.** Standardized genetic (A) and unique environmental (E) path estimates, as well as genetic (rA) and unique environmental (rE) correlations. **S14 Table. Overview of the estimated proportion of variance that is attributable to genetic (A) and unique environmental (E) influences for personality facets included in twin models.**
(ZIP)

## Acknowledgments

We would like to thank the participants from the Norwegian Twin Registry.

## Author contributions

**Conceptualization:** Heidi Marie Umbach Hansen, Espen Røysamb, Olav Mandt Vassend, Nikolai Olavi Czajkowski, Tor Endestad, Jonna Katariina Vuoskoski, Anne Danielsen, Bruno Laeng.

**Data curation:** Heidi Marie Umbach Hansen.

**Formal analysis:** Heidi Marie Umbach Hansen.

**Funding acquisition:** Espen Røysamb, Olav Mandt Vassend, Nikolai Olavi Czajkowski, Anne Danielsen.

**Investigation:** Heidi Marie Umbach Hansen, Espen Røysamb, Olav Mandt Vassend, Bruno Laeng.

**Methodology:** Heidi Marie Umbach Hansen, Espen Røysamb, Olav Mandt Vassend, Nikolai Olavi Czajkowski.

**Project administration:** Heidi Marie Umbach Hansen, Bruno Laeng.

**Supervision:** Espen Røysamb, Bruno Laeng.

**Visualization:** Heidi Marie Umbach Hansen.

**Writing – original draft:** Heidi Marie Umbach Hansen.

**Writing – review & editing:** Heidi Marie Umbach Hansen, Espen Røysamb, Olav Mandt Vassend, Nikolai Olavi Czajkowski, Tor Endestad, Jonna Katariina Vuoskoski, Anne Danielsen, Bruno Laeng.

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
