## [Decision Letter · Decision Letter 0]

4 Jun 2025

Dear Dr. Hansen,

We look forward to receiving your revised manuscript.

Kind regards,

Vilfredo De Pascalis

Academic Editor

PLOS ONE

Journal Requirements:

Additional Editor Comments:

Both reviewers are positive about the manuscript. They reported that the study is well written and the findings are novel. However, they also raised some methodological comments that need to be addressed before the paper is considered for publication.

Reviewers' comments:

Reviewer's Responses to Questions

**Comments to the Author**

1. Is the manuscript technically sound, and do the data support the conclusions?

Reviewer #1: Yes

Reviewer #2: Yes

2. Has the statistical analysis been performed appropriately and rigorously?

Reviewer #1: I Don't Know

Reviewer #2: Yes

3. Have the authors made all data underlying the findings in their manuscript fully available?

Reviewer #1: No

Reviewer #2: No

4. Is the manuscript presented in an intelligible fashion and written in standard English?

Reviewer #1: Yes

Reviewer #2: Yes

Reviewer #1: The manuscript investigates the genetic and environmental underpinnings of individual differences in motivations for music use, as well as their associations with Big five and empathy traits. The topic is novel, manuscript is well structured and clearly presented.

1. The authors mention a high phenotypic and genetic correlations between MM-transcendence and MM- identity, that raise concerns about the discriminant validity of the four MM subscales. Maybe using a confirmatory factor analysis as a follow up analysis might be useful here to access whether these are truly distinct with the sample in mind.

2. The authors can also include supplementary results comparing AE vs ACE models for MM-social which will be very informative.

3. With several personality facets modelled against outcome variables, the analysis involves a multiple comparisons. Although the authors included lower significance threshold, it will be worth correcting for multiple comparisons to show more robust effects (i.e., whether the key results remain the same after correction).

4. The sample predominantly includes middle-aged female Norwegian Twin cohort. Can the authors specify whether age, gender and culture contexts might play a role on personality traits and music use motivations?

Reviewer #2: This was an interesting manuscript which describes the heritability of four dimensions of motivation for music use, and how they relate to various aspects of personality including the Big 5 and Trait Empathy. The manuscript is well written and the findings are novel. I have some comments about the methodology, but I support publication of this paper.

Introduction: The authors refer to some prior studies on links between music and personality. It would be helpful to provide effect sizes, even for just some of the studies summarized. Phenotypic correlations evaluated here are quite small (with some exceptions), so it would be nice to see if these prior “well-established” associations are also modest in size as this would help contextualize those novel findings provided here.

Method/Results: Perhaps my biggest concern is that authors did not focus on the Big 5 personality traits as a whole, but rather focused on 3-4 subscales from 6 different personality dimensions (Big 5 + Empathy). I’m not an expert on the Big 5, but I’m much more familiar with the total scores than these individual subscales and think it is important for the results to also be displayed for the total scales rather than just the subscales (even if this is only a supplementary analysis), as this will make the findings more useful for future researchers.

I would also appreciate some explanation for why it was important to focus on the subscales rather than the total scores (in the intro and/or method).

The correlations among MM subscales are also quite high. Is it worth reporting some of these associations at the aggregate level as well? Again, I appreciate the authors concern to detail regarding whether some subscales may differ from others in their associations with personality (so this is less of a concern than the comment about the Big 5 total scores), but I think a lot of researchers use these measures in aggregate rather than at the subscale level, so having these comparisons would be helpful even if the current results are retained as the primary focus.

I think some sensitivity analyses are warranted for the “Aesthetic Sensitivity” scale. One of the items is “Is fascinated by art, music, or literature” which likely inflated the association between this scale and the music motivation measures. What happens if this item is excluded and you focus just on the other items (about art more broadly)? Do you think future researchers should include this item in the Openness scale when looking at correlations with similar music measures?

95% confidence intervals should be reported throughout the results. I think this is important to contextualize the effect sizes (especially for genetic correlations, which can have very wide SEs when heritabilities of one or both traits are modest).

**Do you want your identity to be public for this peer review?** For information about this choice, including consent withdrawal, please see our Privacy Policy

Reviewer #1: No

Reviewer #2: **Yes: ** Daniel Gustavson

---

## [Author Response · Author response to Decision Letter 1]

4 Jul 2025

Review Comments to the Author

Reviewer #1: The manuscript investigates the genetic and environmental underpinnings of individual differences in motivations for music use, as well as their associations with Big five and empathy traits. The topic is novel, manuscript is well structured and clearly presented.

Response: We thank the Reviewer for the positive evaluation of our work.

Reviewer 1, comment 1. The authors mention a high phenotypic and genetic correlations between MM-transcendence and MM- identity, that raise concerns about the discriminant validity of the four MM subscales. Maybe using a confirmatory factor analysis as a follow up analysis might be useful here to access whether these are truly distinct with the sample in mind.

Response: Thank you for encouraging us to examine this further. As a preliminary examination of this issue, we have included a comparison of a four-factor and one-factor CFA model as a sensitivity analysis, highlighting better fit for the four-factor model. Moreover, we also show that the confidence intervals around the phenotypic correlations do not include 1 and thus corroborate the notion that the four subscales likely reflect distinct yet related concepts. See Methods (lines 398-404), Results (416-421), and Discussion (lines 687-690).

Reviewer 1, comment 2. The authors can also include supplementary results comparing AE vs ACE models for MM-social which will be very informative.

Response: Thank you for this suggestion. We have included this analysis in the Results (lines 440-445), along with a discussion of these results (lines 656-673).

Reviewer1, comment 3. With several personality facets modelled against outcome variables, the analysis involves a multiple comparisons. Although the authors included lower significance threshold, it will be worth correcting for multiple comparisons to show more robust effects (i.e., whether the key results remain the same after correction).

Response: Thank you for the suggestion. We have articulated a rationale behind our approach to alpha adjustments (lines 380-396) and have included adjusted p-values alongside the unadjusted values (as originally reported) in the regression Table and results (lines 525-531).

Reviewer1, comment 4. The sample predominantly includes middle-aged female Norwegian Twin cohort. Can the authors specify whether age, gender and culture contexts might play a role on personality traits and music use motivations?

Response: That is a good point. Although we controlled for age and sex effects, we have highlighted some of these issues in the Limitations section (see lines 837-844).

Reviewer #2: This was an interesting manuscript which describes the heritability of four dimensions of motivation for music use, and how they relate to various aspects of personality including the Big 5 and Trait Empathy. The manuscript is well written and the findings are novel. I have some comments about the methodology, but I support publication of this paper.

Response: We are grateful for the evaluation of our study and report.

Introduction:

Reviewer 2, comment 1: The authors refer to some prior studies on links between music and personality. It would be helpful to provide effect sizes, even for just some of the studies summarized. Phenotypic correlations evaluated here are quite small (with some exceptions), so it would be nice to see if these prior “well-established” associations are also modest in size as this would help contextualize those novel findings provided here.

Response: We agree and have included this point.

Method/Results:

Reviewer 2, comment 2: Perhaps my biggest concern is that authors did not focus on the Big 5 personality traits as a whole, but rather focused on 3-4 subscales from 6 different personality dimensions (Big 5 + Empathy). I’m not an expert on the Big 5, but I’m much more familiar with the total scores than these individual subscales and think it is important for the results to also be displayed for the total scales rather than just the subscales (even if this is only a supplementary analysis), as this will make the findings more useful for future researchers. I would also appreciate some explanation for why it was important to focus on the subscales rather than the total scores (in the intro and/or method).

Response: We agree that we could have provided a clearer rationale for focusing exclusively on facets. We have now added this justification to the introduction, see lines 190-194 and 251-254. Although we understand the concern, we believe it is appropriate to retain the focus on facets. In addition to the reasons outlined in the revised introduction, this approach facilitates interpretation since all predictors are on the same level of the personality hierarchy. Moreover, the associations between the broad Big Five traits and MM dimensions can, to some extent, be inferred from the plot showing all phenotypic correlations between MM dimensions and the personality facets. However, we have included an overview of the phenotypic correlations between the four MM subscales and the broad personality domains in supplementary materials (Table S7). We hope these revisions address the concern.

Reviewer 2, comment 3: The correlations among MM subscales are also quite high. Is it worth reporting some of these associations at the aggregate level as well? Again, I appreciate the authors concern to detail regarding whether some subscales may differ from others in their associations with personality (so this is less of a concern than the comment about the Big 5 total scores), but I think a lot of researchers use these measures in aggregate rather than at the subscale level, so having these comparisons would be helpful even if the current results are retained as the primary focus.

Response: This is a valid point – thank you for raising it. To our knowledge, however, most, if not all, of the studies using the Uses of Music Inventory - which is arguably the most widely used scale in this context - have focused on the distinct dimensions rather than aggregate scores. Moreover, prompted by Reviewer 1, we have included a comparison of a 1-factor vs a 4-factor CFA solution of the MM subscales, showing poorer fit of the 1-factor model. We therefore argue that, until the validity of the aggregate score has been more comprehensively examined, using subscale scores is preferable, as they also offer a more informative and nuanced understanding.

Reviewer 2, comment 4: I think some sensitivity analyses are warranted for the “Aesthetic Sensitivity” scale. One of the items is “Is fascinated by art, music, or literature” which likely inflated the association between this scale and the music motivation measures. What happens if this item is excluded and you focus just on the other items (about art more broadly)? Do you think future researchers should include this item in the Openness scale when looking at correlations with similar music measures?

Response: Thank you for the excellent point. We agree that this is an important concern, and we have now included a sensitivity analysis excluding the item “Is fascinated by art, music, or literature” from the Aesthetic Sensitivity facet (see Methods section, lines 404-409; Results, lines 501-505; Discussion, lines 725-731). Notably, we conducted a similar analysis in our previous publication (https://www.nature.com/articles/s41598-025-95661-z), where we also found that the association between Aesthetic Sensitivity and musical sensibility remained substantial and statistically significant even when this item was excluded. Similar results were reported in a study by Greenberg et al (see: https://doi.org/10.1016/j.jrp.2015.06.002). The present results replicate this pattern by showing that the associations remain robust, and Aesthetic Sensitivity continues to emerge as the strongest predictor across all MM subscales. Combined, these findings would seem to suggest that the associations are not solely driven by this particular item. Nevertheless, we agree that including such sensitivity checks is important for assessing potential self-report bias arising from conceptual overlap.

Reviewer 2, comment 5: 95% confidence intervals should be reported throughout the results. I think this is important to contextualize the effect sizes (especially for genetic correlations, which can have very wide SEs when heritabilities of one or both traits are modest).

Response: We agree and have added this.

---

## [Decision Letter · Decision Letter 1]

22 Jul 2025

Dimensions of music use motivations: Genetic and environmental underpinnings, and associations with Big Five- and empathy traits

PONE-D-25-13148R1

Dear Dr. Hansen,

We’re pleased to inform you that your manuscript has been judged scientifically suitable for publication and will be formally accepted for publication once it meets all outstanding technical requirements.

Kind regards,

Vilfredo De Pascalis

Academic Editor

PLOS ONE

Additional Editor Comments (optional):

The authors substantially addressed all the comments raised by both reviewers. Their response to Comment 2 raised by Reviewer 2 is only partially satisfactory. As reported by Reviewer 1, the paper would be more useful to the field if total scores for the Big 5 traits were also reported, in addition to the facet-level effects it focuses on here. However, the manuscript maintains scientific validity and can be accepted for publication in its current form.

Reviewers' comments:

Reviewer's Responses to Questions

**Comments to the Author**

Reviewer #2: All comments have been addressed

2. Is the manuscript technically sound, and do the data support the conclusions?

Reviewer #2: Yes

3. Has the statistical analysis been performed appropriately and rigorously?

Reviewer #2: Yes

4. Have the authors made all data underlying the findings in their manuscript fully available?

Reviewer #2: No

5. Is the manuscript presented in an intelligible fashion and written in standard English?

Reviewer #2: Yes

Reviewer #2: The authors did a good job of responding to the initial feedback. I disagree with their response to Reviewer 2, comment 2 and still think the paper would be more useful to the field if total scores for the Big 5 traits (e.g., extraversion) were reported in addition the facet-level effects they focus on here. However, this should not hold up publication of the paper.

**Do you want your identity to be public for this peer review?** For information about this choice, including consent withdrawal, please see our Privacy Policy

Reviewer #2: No

---

## [Editor Report · Acceptance letter]

PONE-D-25-13148R1

PLOS ONE

Dear Dr. Hansen,

I'm pleased to inform you that your manuscript has been deemed suitable for publication in PLOS ONE. Congratulations! Your manuscript is now being handed over to our production team.

Kind regards,

on behalf of

Prof. Vilfredo De Pascalis

Academic Editor

PLOS ONE